# Modeling the Heating Dynamics of a Semiconductor Bridge Initiator with Deep Neural Network

**DOI:** 10.3390/mi13101611

**Published:** 2022-09-27

**Authors:** Jianbing Xu, Jimin Tan, Hanshi Li, Yinghua Ye, Di Chen

**Affiliations:** 1The Future Laboratory, Tsinghua University, Beijing 100084, China; 2Academy of Arts & Design, Tsinghua University, Beijing 100084, China; 3Center for Data Science, New York University, New York, NY 10012, USA; 4Department of Mechanical Engineering and Materials Science, Yale University, New Haven, CT 06520, USA; 5School of Chemistry and Chemical Engineering, Nanjing University of Science and Technology, Nanjing 210094, China

**Keywords:** semiconductor bridge (SCB), deep learning (DL), multi-physical field simulation (MPS), initiator

## Abstract

A semiconductor bridge (SCB) is an ignition device that provides a safe and efficient method widely used in civilian and military fields. The heating process of an SCB under electrical stimulation has a wide range of applications owing to its unique energy release process. However, the temperature variation of an SCB is challenging to obtain, both experimentally because of the rapid reaction on a microscale and with simulation due to its high demand in nonlinear calculations. In this study, we propose deep learning (DL) approach to study the electrothermal-coupled multi-physical heating process of the SCB initiator. We generated training data with multi-physics simulation (MPS), producing surface temperature distributions of SCBs under different voltages. The model was then trained with partial data in this database and evaluated on a separate test set. A generative adversarial network (GAN) with a customized loss function was used for modeling point-wise temperature dynamics. In the test set, our proposed method can predict the temperature distribution of an SCB under different voltages with high accuracy of over 0.9 during the heating process. We reduced the computation time by several orders of magnitude by replacing MPS with a deep neural network.

## 1. Introduction

A semiconductor bridge (SCB), which can generate heat and plasma in a short time to ignite subsequent charge, has great ignition ability compared with traditional bridge-wire pyrotechnics [1,2]. However, because of the heavily doped polysilicon fabrication, SCBs possess a negative temperature coefficient. When the pulse current flows through the polysilicon, SCBs can quickly reach maximum power, resulting in electrical explosions [3]. Therefore, investigating the heating process of SCBs under electrical stimulation could yield further insights into the dynamics, aiding the design choices of SCBs.

Many empirical formulations were proposed in previous studies on the electric explosion performance of SCBs, such as critical explosion time and energy [4,5]. Combing the oscilloscope and high-speed camera analysis, the heating process of SCBs is divided into four steps: warming, melting, gasification, and ionization [6,7]. In addition, the heat transfer model of SCBs is established under the assumption that SCBs have a monolithic heating source [8,9]. However, these models cannot reveal the actual heating dynamics of the SCB since the microstructure of the SCB is not considered. When more fine structures of SCBs are considered (width, length, thickness, angle), a multi-physics simulation (MPS) based on finite element analysis is required [5,7,8] because SCBs are highly nonlinear with various heating processes. The MPS has shown high accuracy in the performance analysis, but it remains too complex for online prediction and optimization. Moreover, the number of computational grids needs to be increased to get more accurate results, increasing the computational burden even further.

With the rapid development in the machine learning field, it has been shown that deep neural networks can solve the problem of the insufficient computing power of the multi-physical field model [10,11]. Furthermore, as a data-driven method, deep neural networks can effectively combine experimental data for self-learning and updating [12], providing a new way for solving and optimizing complex models. The performance of replacing those complex models with deep neural networks has been validated [13,14,15]. In addition, deep learning has been widely applied in material science subfields such as material development [11,16] and battery life detection [17,18].

This data-driven approach significantly reduces the calculation time compared to the MPS models. Therefore, we propose a deep neural network to study the electrothermal-coupled multi-physical heating process of the SCB initiator. The training data set was generated by solving the physical model with COMSOL Multiphysics, a finite element analysis software that requires much less time and cost than an experimental method. Moreover, this dataset can provide detailed temperature information of SCBs, which is difficult to collect from experiments. And a Generative Adversarial Network (GAN) framework was used to train the model for predicting point-wise temperature variations.

## 2. Experiments

### 2.1. Electrothermal-Coupled Multi-Physical Field Simulation of SCBs

The heating process of the semiconductor bridge initiator is a complex physical and chemical reaction consisting of melting, gasification, and ionizing polysilicon material. The polysilicon semiconductor can obtain high power with outer electricity stimulus and make the silicon vapor ionize due to the negative temperature coefficient characteristic. Thus, it is necessary to consider the change in the resistivity of polysilicon material at different temperatures. When SCB is heated under electricity, its resistance will be variable since the resistivity of polysilicon material is related to temperature [2]. This, in turn, results in a change in the heat generation rate of the SCB. Therefore, the heating process of SCBs under electrical is the simultaneous action process of electric field and thermal field, which can be expressed as Equation (1).
(1)$$\left\{ \begin{array}{l} \rho_{\mathrm{SCB}}C_{\mathrm{SCB}}\frac{\partial T_{\mathrm{SCB}}}{\partial t}=\nabla •\left(\lambda_{\mathrm{SCB}}\nabla T_{\mathrm{SCB}}\right)+P_{\mathrm{SCB}}\\ P_{\mathrm{SCB}}=J_{\mathrm{SCB}}•E_{\mathrm{SCB}}\\ J=\frac{1}{\rho_{0\;\mathrm{SCB}}}•E_{\mathrm{SCB}}\\ E_{\mathrm{SCB}}=\nabla V_{\mathrm{SCB}} \end{array}\right.$$

Where *ρ*_SCB_ (kg•m^−3^) is the density of SCBs, *C*_SCB_ (J•kg^−1^•K^−1^) is the specific heat of SCBs, *λ*_SCB_ (W•m^−1^•K^−1^) is the heat conductivity of SCBs, are listed in Table 1. *P*_SCB_ (W) is the power of SCBs, *J*_SCB_ (A•m^−2^) is the current density of SCBs, *E*_SCB_ (V•m^−1^) is the electric field strength of SCBs, *V*_SCB_ (V) is the applied electric potential of SCBs, *ρ*_0 SCB_ (Ω·m) is the resistivity of SCBs [19]. When we conduct multi-physical field coupling simulation, it is necessary to consider the temperature dependence of the parameters of the research model. However, in this study, SCB is mainly heated by its resistance effect, and the change of resistivity will affect its heating power, so the temperature dependence of resistivity is primarily considered. Here, the resistivity of SCBs is the temperature function, shown in Equation (2).
(2)$$\rho_{0}(T)=\left\{ \begin{array}{ll} 5.0009\times 10^{4} & T<298\\ 1/(2\times 10^{-5}*(1+1.25\times 10^{-3}*(T-298.15))) & 298<T<1684\\ 1.23\times 10^{6}-2.7*(T-1684) & 1684<T<2880\\ 1.2268\times 10^{6} & T>2880 \end{array}\right.$$

Figure 1a shows the SCB image consisting of Si/SiO_2_ substrate, SCB layer, and Au pad on both sides. The area of the exposed SCB is 380 µm × 80 µm with a double V shape (angle of 90 °). The size of the Au pad is 1000 µm × 710 µm. The commercial software COMSOL Multiphysics, which provides finite element analysis of multi-physics, including electricity and heat transfer, was used to calculate the heating process of SCBs. In this study, our primary purpose is to verify the feasibility of machine learning methods for SCB research. Therefore, for the data set of the model, we obtain the extensive data set by the simple multi-physical field coupling model. A simplified two-dimensional computational model was adopted to speed up the calculation and get more data, as shown in Figure 1b. Figure 1c shows the finite element mesh division, and the typical result shows in Figure 1d.

### 2.2. Data Generation and Model Training

By deliberately varying the voltages applied to SCB, we generated a dataset that captures a series of surface temperature variations during 10 μs with a 0.1 μs time interval. Thus, the voltage was applied from 0 V to 25 V with a 0.1 V increment. Figure 2a shows how the training dataset is extracted from the simulation. The dataset contained 250 voltage results with 100 temperature distributions at different times per voltage. Since the temperature distribution of SCBs was axisymmetric, a quarter of the temperature data was selected as our raw data, as shown on the left of Figure 2a. Figure 2c shows the surface temperature distribution of raw data. Each sample contains 500 × 750 points temperature, with 1 µm apart from each temperature point. Since the simulation result was symmetrical on the central horizontal and vertical axis, we picked a 256 by 256 window adjacent to the two axes in the first quadrant as experimental examples during modeling. This selection reduced the computational power required for model training and didn’t undermine model performance because it includes the area of interest. Only a part of the sequence data is selected to train the model, as shown on the left of Figure 2b. To train the deep learning model, we choose 100 data at 1μs as input data and another 100 data at 10 μs as target data, as shown in Figure 2d. The data points before SCB melting are used for modeling to simplify the learning setup. We expect that from the SCB state at 1 μs, the model can deduce the corresponding state at 10 μs under constant voltage. Therefore, we leave out specific voltages for the test set. Specifically, we selected a wide range of data at 1 V, 5 V, 10 V, and 20 V to test the performance of our model at different voltages. The final model is trained for 328 epochs on the training set and tested on four voltages to evaluate its performance.

### 2.3. Machine Learning Framework: Conditional Generative Adversarial Network

Since we want to model point-wise continuous temperature dynamics in the SCB system, traditional machine learning approaches such as linear regression and SVM failed. And they could not model the complex interdependency between different points on SCB, including the spatial relationship at neighboring elements. Deep learning has shown superiority over traditional algorithms in learning information from raw signals and integrating complex dependencies through its hierarchical structure. Specifically, convolutional neural networks (CNN) can integrate dependence between spatially close signals through their convolution kernels, making it widely applicable for a signal with spatial dimensions, including natural images. Similarly, CNN can be leveraged to model the spatial heat distribution in SCB and is used as a modeling backbone in our proposed model.

CNN trained with MSE (mean squared error) loss tends to smooth the output. Since this application requires high numerical accuracy, we consider the smoothing effect a compromise to model performance. To alleviate this issue, we built a conditional variant of generating an adversarial network (GAN) on top of the CNN backbones to capture the nuanced variations during the heat process. As shown in Figure 3, conditional GAN consists of a generator network (G) that produces an element-wise temperature distribution from a conditioned variable, the input SCB temperature distribution. A discriminator network (D) also takes the conditional and distinguishes the actual distributions from the generated distributions. They reach an equilibrium by constructing G and D to optimize a minimax loss function [20]. Conditioned G learns the distribution characteristics based on the conditional in the game process. Thus, given a conditional input, G captures the distribution of actual data samples and generates new data samples.

We implemented a U-Net with convolutional layers as our generative model because the skip connection improves reconstruction quality [21]. The loss function is a standard GAN loss function with our custom ratio loss at the end [22]. The ratio loss is crucial for reaching a stable prediction since the spectrum of temperatures range can be wide depending on different voltages.
(3)$$L_{cGAN}(G,D)=E_{x}[\log(D(x,y))]+E_{z}[\log(1-D(x,G(x,z))+\gamma (\frac{G(x,z)}{y}-1)]$$

The GAN loss consists of three components: the discriminator loss (first term), generator loss (second term), and our customized ratio loss (last term). Since the surface temperature does not follow a Gaussian distribution, the prediction tends to overshoot. To alleviate this problem, we added the ratio loss to add an extra penalty for the generator if it produces inaccurate heat distributions in low-temperature regions. The parameter γ is a hyperparameter that adjusts the degree of correction and is set to 1 for this study.

## 3. Results and Discussion

### 3.1. Typical Simulation Result of SCBs

To research the heating process of SCBs, the typical surface temperature of SCBs simulated with COMSOL was analyzed. The surface temperature distribution of SCBs at 1 μs, 4 μs, 7 μs, and 10 μs under 20 V is shown in Figure 4, and the heating process is shown in Appendix A. The temperature of the double V-shaped SCB raised fastest at the edge of the sharp V-shaped corner (Figure 4). Temperature increases slowly for regions that are further away from the V-shaped corner. The highest temperature was first observed at the edge of the sharp V-shaped corner. Then, along the direction of the narrowest part of the SCB, the hot spot spread from both ends to the middle. The hot spot extends from the narrowest point to both ends in the width dimension. Because of the sharp V-shaped corners, the current density was the highest at the sharp V-shaped corners. Therefore, it was easier to generate hot spots at sharp corners in SCB. This is consistent with our previous result that the sharp V-shaped corner has the largest electric explosion area [23,24].

### 3.2. Comparison and Analysis of the Prediction for the Heating Process of SCBs

Figure 5 shows the surface temperature of SCBs at 10 μs under 5 V, 10 V, 15 V, and 20 V as predicted by the proposed model, and the deep learning process is shown in Appendix A. The first column graph in Figure 5 shows the surface temperature of SCBs at 1 μs under 5 V, 10 V, 15 V, and 20 V, which were taken as input for the model. The second column shows the model prediction of the SCB surface temperature at 10 μs (Figure 5). The third column shows the ground truth of the surface temperature of SCBs at 10 μs from COMSOL simulation. Comparing the prediction result and the ground truth (the second column graph and the third column graph in Figure 5), the absolute error is minimal, demonstrating the effectiveness of the proposed model in predicting heating dynamics (fourth column, Figure 5). At all voltages, the mean error was under 25 K across the SCB surface with a standard deviation of less than 15 K. The low average error rate means the model has high reconstruction quality. We examined the prediction on high-temperature regions to validate problem-specific model performance. The maximum absolute error of temperature prediction for all four validation cases was 107.91 K which occurred at 10 V. At 20 V, the absolute max difference was 58.59 K within a temperature range between 320 K and 1035 K, which achieved an accuracy of over 90%. It is worth noting that the metrics are calculated on the entire surface of SCBs. Thus, every point on the surface has a lower error than the maximum error, demonstrating the consistency of the generated temperature distributions.

We can see that predictions at lower voltages were more likely to overshoot in high-temperature regions (Figure 6), and as the voltage increased, the model prediction was more accurate. The extremely high maximum temperature can explain this phenomenon at higher voltages (Over 1000 K max at 20 V). The original loss function was based on L2 loss, which had greater penalization for a larger deviation from the mean value of the temperature distribution. To minimize L2 loss, the model had to fit extremely high-temperature regions at high voltage correctly and thus resulting in a universal overshoot at lower voltages. We introduced ratio loss to mitigate this problem partially. However, in this specific set of SCBs, we are more interested in the maximum temperature value. Dialing the weight of ratio loss too far can cause a deviation of high-temperature regions under high voltage. Thus, we controlled the importance of ratio loss to improve overall reconstruction quality without losing model performance in critical regions.

### 3.3. The Comparison of Computation Ability between DL Model and Physical Model

The run time for our model was much shorter than the simulation on COMSOL Multiphysics software. During inference, we extracted the generator and fed it an image to predict an outcome. Depending on hardware specifications, the process usually took less than 1 s, while the simulation time with COMSOL Multiphysics was 11 s. The computation time of the DL model was reduced by order of magnitude compared with a physical model. Table 2 summarizes the Computation ability between the DL model and the physical model. The physical model was computed on Intel i7-9750 CPU (2.6 GHz), while the DL model was computed on RTX 2080 Ti GPU. Compared to a CPU based on serial computation, GPU was based on parallel computation, which made its computation time far less than ordinary CPU.

## 4. Conclusions

In this study, we proposed a deep neural network architecture to predict the surface temperature of SCBs with different input energy. A database consisting of 250 × 100 surface temperature distribution figures with 100 time points (0.1 µs to 10 µs with the interval of 0.1 µs) and 250 voltage points (0.1 V to 25 V with the interval of 0.1 V) are generated with MPS by COMSOL Multiphysics, in which the electrical and heat transfer were fully considered. One hundred data sets at 1 µs with different voltages were selected as input data. Another 100 sets of data at 10 μs were chosen as target data to train the deep neural network. We implement a custom loss function besides the standard GAN loss to adjust for the overshooting error caused by extremely high temperatures on the SCB surface and enhance prediction quality at lower temperature regions. The surface temperature in the SCB was predicted using the generator from GAN under different operating voltages, showing high consistency with the multi-physics simulation. In conclusion, we develop a convolutional neural network trained with adversarial and customized ratio loss that can accurately predict the temperature distribution of SCBs with an accuracy of over 90%. Our implementation also reduces the calculation time by an order of magnitude.

Although our study did not involve the nonlinear part after SCB melting, the results show that the deep neural network can solve the electrothermal problem with multiple physical field coupling. The ignition performance of the SCB depends on the heating situation of the whole bridge area. Compared to MPS, our model can quickly predict the temperature profile of the entire bridge area of the SCB. Therefore, this study is helpful in rapidly evaluating the firing performance of the SCB. Thus, in the future, our model could be trained with data from an experiment and achieve a similar level of performance. We expect the model to be further generalizable to different parameters combination of the SCB, reducing development costs and the need for extensive experiments.

## Figures and Tables

**Figure 1 micromachines-13-01611-f001:**
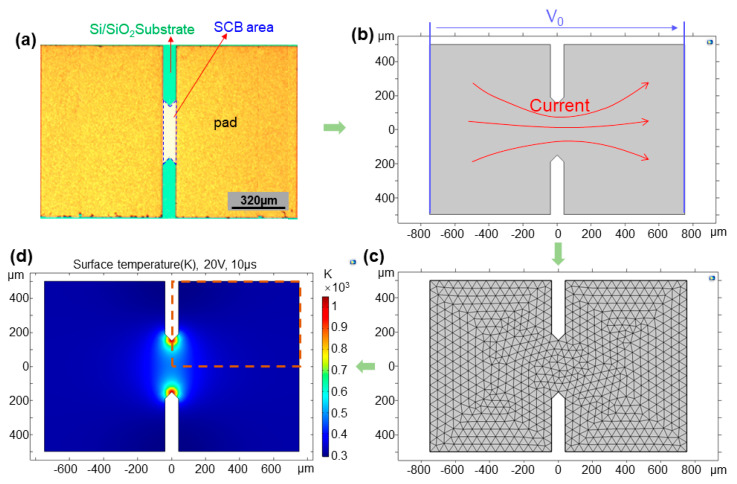
(**a**) Photograph of the SCB initiator, the SCB area is 380 µm × 80 µm with Au/Ti pad on both sides, (**b**) 2D physical simulation of SCBs, a voltage is applied on SCB, (**c**) the finite element mesh division, (**d**) typical surface temperature simulation result of SCBs.

**Figure 2 micromachines-13-01611-f002:**
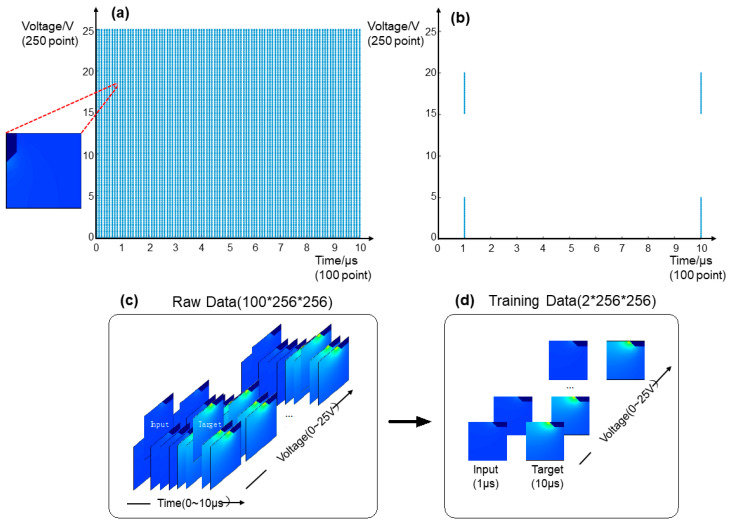
(**a**) A schematic of the raw dataset of a simulation result. The dataset contained 250 voltage results with 100 temperature distributions at different times per voltage (**b**) A schematic of training data of simulation result, 100 data at 1 μs as input data, and another 100 data at 10 μs as target data (**c**) surface temperature distribution of raw data, (**d**) surface temperature distribution of training data.

**Figure 3 micromachines-13-01611-f003:**
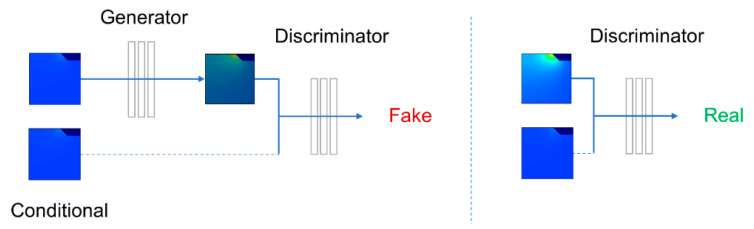
Training and testing flowchart of deep learning model, a conditional variant of Generate adversarial network (GAN) on top of the CNN backbones was adopted to capture the nuanced variations during the heat process.

**Figure 4 micromachines-13-01611-f004:**
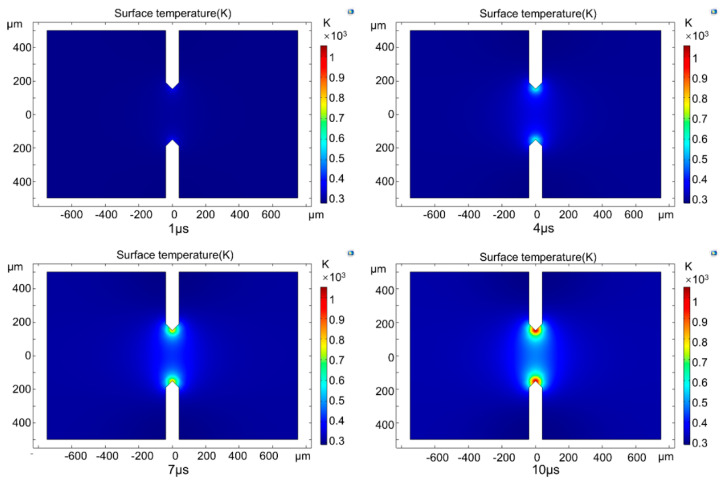
The typical surface temperature of SCBs at 1 μs, 4 μs, 7 μs, and 10 μs under 20 V, the V-shaped corner of SCBs has a maximum heating rate.

**Figure 5 micromachines-13-01611-f005:**
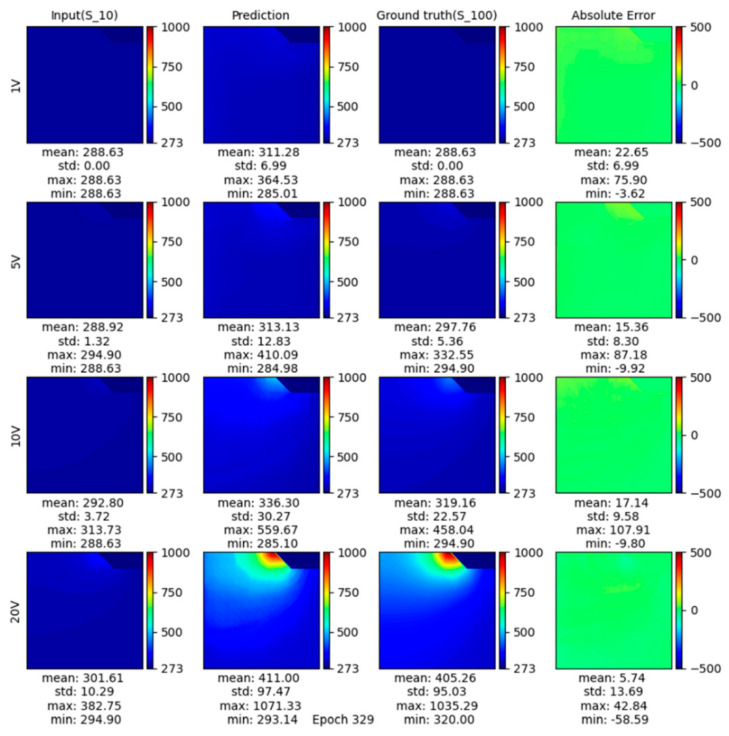
Deep learning provides temperature predictions on the SCB region at 10 μs at 1 V, 5 V, 10 V, and 20 V. The first column is input at a different voltage at 1 μs. The second column is output prediction by the DL model at different voltages at 10 μs, the third column is ground truth, and the fourth column is the absolute error about ground truth and prediction.

**Figure 6 micromachines-13-01611-f006:**
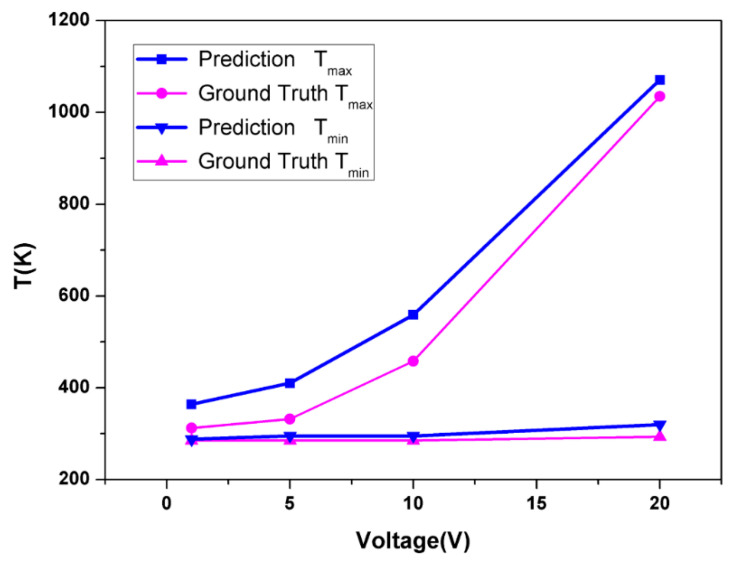
Comparison of predicted and calculated values of maximum and minimum temperatures at different voltages.

**Table 1 micromachines-13-01611-t001:** Polysilicon properties.

*Ρ* (×10^3^ kg•m^−3^)	*λ* (W•m^−1^•K^−1^)	*C* (J•kg^−1^•K^−1^)
2.33	149	713.8

**Table 2 micromachines-13-01611-t002:** Computation ability between DL model and physical model.

Computation Model	Computing Hardware	Computing Software	Computation Time (s)
DL model	RTX 2080 Ti GPU	Python	<1
Physical model	Intel i7-9750 CPU 2.6 GHz	COMSOL Multiphysics 5.4	11

## Data Availability

The data that support the findings of this study are available from the corresponding author upon reasonable request.

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
