# Peer review of "Modeling the Heating Dynamics of a Semiconductor Bridge Initiator with Deep Neural Network"

_micromachines, 2022, doi:10.3390/mi13101611_

Round 1
Reviewer 1 Report
The ignition ability of semiconductor bridge is largely determined by its heating process. It is challenge to quickly predict the heating process in the whole area for SCB research since the heating process takes place in a very short time (micro second) and tiny area (micro meter). In this article, the authors firstly propose a deep neural network to study the electrothermal-coupled multi-physical heating process of the SCB initiator. A convolutional neural network trained with adversarial and customized ratio loss can accurately predict the temperature distribution of SCB. It reduces the calculation time by an order of magnitude. This work opens a new method on the research of micro ignition device. However, there are still some deficiencies in the current version that need to be minor revised before the manuscript could be considered for publication in Micromachines.
Specific comments:
1. The authors generate training data of SCB by electrothermal-coupled multi-physical field simulation. The relationship between resistivity and temperature is considered. Why is the temperature dependence of other parameters not considered, such as specific heat, heat conductivity?
2. In Figure 1(c), the finite element mesh division were evenly divided when they meshed. No mesh refinement was done in a specific. Does this affect the accuracy of the simulation results? Does this have any influence on subsequent deep learning simulations?
3. Figure1 (d) Figure 4(d) are the simulation result under the same condition of 5V/10μs.The temperature distributions of this two Figures is inconsistent. Why? Please confirm the simulation condition of these figures.
4. There are many minor mistakes. In page 1, line 27, “ESCB” →“SCB”; In page 7, line 211, “Fig. 6” →“Figure 6”; In page 8, line 228, “Table â…¡” →“Table 2”. The author needs to revise it and check the manuscript carefully again.
5. Please check the reference style of [2], [5], [6], [18], [22]. The reference section should be carefully checked and modified carefully as well.
Author Response
Response to Reviewer 1 Comments
To Reviewer 1:
Thank you very much for your kindly comments on our paper submitted to Micromachines (Manuscript ID: micromachines-1939309, Title: Modeling the heating dynamics of a semiconductor bridge initiator with deep neural network). We apologized for our carelessness. We have checked the manuscript and revised some errors. The replies to your comments are as follows:
Point 1: The authors generate training data of SCB by electrothermal-coupled multi-physical field simulation. The relationship between resistivity and temperature is considered. Why is the temperature dependence of other parameters not considered, such as specific heat, heat conductivity?
Response 1: Thank you for your comment. This is a good question. When we conduct multi-physical field coupling simulation, it is indeed necessary to consider the temperature dependence of the parameters of the research model. However, in this study, SCB is mainly heated by its resistance effect, and the change of resistivity will affect its heating power, so the temperature dependence of resistivity is mainly considered. As suggested by the reviewer, we have added the explanation in the revised manuscript. (see change in Page 2, line 84-88)
Point 2: In Figure 1(c), the finite element mesh division were evenly divided when they meshed. No mesh refinement was done in a specific. Does this affect the accuracy of the simulation results? Does this have any influence on subsequent deep learning simulations?
Response 2: Thank you for your good comment. This is a good question. In this study, our main purpose is to verify the feasibility of machine learning methods for SCB research. Therefore, for the data set of the model, we obtain the big data set by the simple multi-physical field coupling model. Our research focuses on the training of the late deep learning model, so the multi-physical field data set will not affect the late deep learning process. And the results show that the deep neural network can solve the electrothermal problem with multiple physical field coupling. As suggested by the reviewer, we have added the explanation in the revised manuscript. (see change in Page 3, line 99-102)
Point 3: Figure1 (d) Figure 4(d) are the simulation result under the same condition of 5V/10μs.The temperature distributions of this two Figures is inconsistent. Why? Please confirm the simulation condition of these figures.
Response 3: Thank you for your good comment. Figure1 (d) Figure 4(d) show the simulated temperature distribution of the bridge area at 20V, but we don't have a uniform scale bar. We have modified this section in the revised manuscript. (see change in Page 3, line 91)
Point 4: There are many minor mistakes. In page 1, line 27, “ESCB” →“SCB”; In page 7, line 211, “Fig. 6” →“Figure 6”; In page 8, line 228, “Table â…¡” →“Table 2”. The author needs to revise it and check the manuscript carefully again.
Response 4: Thank you for your kind reminding. We have revised these minor mistakes and checked the manuscript again to make them better.. (see change in Page 1, line 27; Page 3, line 93, Page 3, line 102; Page 5, line 176; Page 6, line 187; Page 7, line 217; Page 8, line 234)
Point 5: Please check the reference style of [2], [5], [6], [18], [22]. The reference section should be carefully checked and modified carefully as well.
Response 5: Thank you for your good comment. Thank you for your comment. We have carefully checked all references and have modified this section in the revised manuscript. (see change in References, Page10, line 278-325)

Reviewer 2 Report
The authors present an exciting paper where they propose deep learning (DL) approach to study the electrothermal-coupled multiphysical heating process of the SCB initiator. They generated training data with multi-physics simulation (MPS), producing surface temperature distributions of SCB under different voltages. The model was then trained with partial data in this database and evaluated on a separate test set. A generative adversarial network (GAN) with a customized loss function was used for modelling point-wise temperature dynamics. The manuscript’s results are reproducible based on the details given in the methods section. The manuscript is well written and should greatly interest the readers. The remarks are to explain more about deep learning and why they choose this technique, and the conclusion should mention more about their future work.
Author Response
Response to Reviewer 2 Comments
To Reviewer 2:
Thank you very much for your kindly comments on our paper submitted to Micromachines (Manuscript ID: micromachines-1939309, Title: Modeling the heating dynamics of a semiconductor bridge initiator with deep neural network). Your suggestions are very useful to us, and thank you for your recognition of our work. We have checked the manuscript again and revised it to meet the publication standards of Micromachines.
